

# Transcriptome differential expression analysis of defoliation of two different lemon varieties

Meichao Dong[1], Tuo Yin[2], Junyan Gao[1], Hanyao Zhang[2], Fan Yang[1], Shaohua Wang[1], Chunrui Long[1], Xiaomeng Fu[1], Hongming Liu[1], Lina Guo[1] and Dongguo Zhou[1]

[1] Institute of Tropical and Subtropical Cash Crops, Yunnan Academy of Agricultural Sciences, Baoshan, China

[2] The Key Laboratory of Biodiversity Conservation of Southwest China, National Forestry and Grassland Administration, College of Forestry, Southwest Forestry University, Kunming, China

## ABSTRACT

'Allen Eureka' is a bud variety of Eureka lemon with excellent fruiting traits. However, it suffers from severe winter defoliation that leads to a large loss of organic nutrients and seriously affects the tree's growth and development as well as the yield of the following year, and the mechanism of its response to defoliation is still unclear. In order to investigate the molecular regulatory mechanisms of different leaf abscission periods in lemon, two lemon cultivars ('Allen Eureka' and 'Yunning No. 1') with different defoliation traits were used as materials. The petiole abscission zone (AZ) was collected at three different defoliation stages, namely, the pre-defoliation stage (CQ), the mid-defoliation stage (CZ), and the post-defoliation stage (CH). Transcriptome sequencing was performed to analyze the gene expression differences between these two cultivars. A total of 898, 4,856, and 3,126 differentially expressed genes (DEGs) were obtained in CQ, CZ, and CH, respectively, and the number of DEGs in CZ was the largest. GO analysis revealed that the DEGs between the two cultivars were mainly enriched in processes related to oxidoreductase, hydrolase, DNA binding transcription factor, and transcription regulator activity in the defoliation stages. KEGG analysis showed that the DEGs were concentrated in CZ and involved plant hormone signal transduction, phenylpropanoid biosynthesis, glutathione metabolism, and alpha-linolenic acid metabolism. The expression trends of some DEGs suggested their roles in regulating defoliation in lemon. Eight gene families were obtained by combining DEG clustering analysis and weighted gene co-expression network analysis (WGCNA), including β-glucosidase, AUX/IAA, SAUR, GH3, POD, and WRKY, suggesting that these genes may be involved in the regulation of lemon leaf abscission. The above conclusions enrich the research related to lemon leaf abscission and provide reliable data for the screening of lemon defoliation candidate genes and analysis of defoliation pathways.

Corresponding author
Dongguo Zhou, rjszdg@yaas.org.cn

## INTRODUCTION

Lemons (*Citrus limon* (L.) Burm. F.) are globally recognized for their fruits' health, nutrition and medicinal value; being rich in vitamins, citric acid, flavonoids, and a variety of minerals and trace elements (*González-Molina et al., 2010*); and blood lipid-lowering, uric acid-lowering, antioxidant, and anti-cancer effects (*Benavente-García & Castillo, 2008*). As living standards have improved, the demand for lemons has increased significantly, and the development of lemons has become increasingly rapid. Lemons are tropical and subtropical evergreen tree species. However, the production of 'Allen Eureka' lemon in Yunnan, China has been found to have abnormal defoliation in the winter, which has seriously affected the growth and development of the tree and its yield.

Abscission is a physiological process prevalent in living organisms' leaves (*Liao et al., 2023*), flowers (*Lu et al., 2023*), fruits (*Yan et al., 2021*), roots (*Zhu et al., 2019*), and other organs. The area where organ abscission occurs is called the abscission zone (AZ). AZ differentiation forms abscission layers, which are closely related to abscission. Currently, the molecular mechanism of floral organ abscission has been thoroughly studied in *Arabidopsis thaliana*, where it consists of four stages: formation of the AZ, perception and response to abscission signals, enzymatic detachment of the cells in the AZ, and formation of protective layer cells (*Kim, 2014*).

The development and functional exercise of the abscission layer is a complex and precise process in which multiple enzymes (*Tong et al., 2020*; *Wang et al., 2023*), hormones (*Kućko, Wilmowicz & Ostrowski, 2019*; *Li et al., 2023*), and genes (*Wang et al., 2021*; *Ma et al., 2022*) are involved in regulation. Abscission is a physiological process associated with stress response and plant organ senescence. It is generally considered to be the result of stress (*e.g.*, drought, low light, extreme temperatures, or pathogen infestation) at the distal end of the rachis (near the base of the petiole) that causes the plant body to enter senescence and produce an abscission signal that is transmitted to the AZ (*Cheng et al., 2022*; *Bar-Dror et al., 2011*).

The regulation of abscission by phytohormones is universal. Almost all hormones regulate abscission to some extent, with phytohormones such as ethylene (ACC), abscisic acid (ABA), jasmonic acid (JA), and cytokinin (CTK) acting as accelerating signals for abscission to promote abscission, and gibberellin (GA), growth hormone (Auxin), polyamines, and oleoresin stanol hormones acting as inhibiting signals for abscission to inhibit abscission (*Li et al., 2023*; *Parra-Lobato & Gomez-Jimenez, 2011*). However, plant organ abscission is controlled by a combination of hormones.

The direct cause of plant organ abscission is the degradation of the cell wall, which in turn is due to several associated cellulases (*Wang et al., 2023*; *Li et al., 2019*), pectinases (*Bai et al., 2023*), peroxidases (*Djanaguiraman et al., 2010*), polygalacturonases (*Hu et al., 2023*; *He et al., 2023*; *Bonghi et al., 1992*), xyloglucan endoglucan transglucosylase/hydrolyzing enzymes (*Wang et al., 2022*; *Niederhuth, Patharkar & Walker, 2013*), and extender proteins that lead to the disruption of cell-to-cell adhesion (*Patharkar & Walker, 2018*; *Botton & Ruperti, 2019*).

Apart from *Arabidopsis*, the signaling pathway of leaf abscission has been reported less in other plants, but some cell wall hydrolases and signaling genes have been isolated and shown to be involved in the regulation of organ abscission. For example, citrus leaves were treated with ethylene glycol, and transcriptome sequencing was performed on the treated abscission sites. Analysis of ribonucleic acid (RNA)-Seq results showed that cell wall-modifying enzyme genes, stress-related genes, pathogen-related genes, MAPK kinase-related genes, transcription factors, ethylene biosynthesis, and signaling genes were significantly expressed in the exfoliated position (*Agustí et al., 2008*, *2009*). CsPLDα1 (phospholipase D) and CsPLDγ1 are involved in citrus abscission through a hormone-dependent signaling pathway. CsPLDα1 and CsPLDγ1 encode phospholipase D, and their expression levels are affected by ethylene, which may be dependent on the ethylene signaling pathway for the regulation of citrus abscission (*Malladi & Burns, 2008*).

Defoliation is a manifestation of damage in plants subjected to abiotic stresses such as low temperature and drought, as well as a defense mechanism to reduce water loss and protect the tree; however, its intrinsic mechanism remains unclear. In order to deeply study the molecular biological mechanism of lemon defoliation, the petiole AZ of two lemon varieties with different defoliation traits, 'Yunning No. 1'/*Poncirus trifoliate* (L.) Raf and 'Allen Eureka'/*Poncirus trifoliate* (L.) Raf, were selected as the materials, and their enzyme activities, hormone contents, and transcriptome sequencing were analyzed. We compared changes in physiological indexes and the functional analysis of differential genes of the two lemon varieties at different periods of defoliation. The biological differences between the two lemon varieties in the process of defoliation were clarified, filling in the gap of genes related to the regulation of lemon defoliation, and providing theoretical basis for further excavation of key genes regulating lemon defoliation.

# MATERIALS AND METHODS

## Experimental materials

Previously, our group conducted observation records and research on two lemon varieties ('Allen Eureka' and 'Yunning No. 1') for three consecutive years (Figs.1A and 1B). As shown in Fig. 1, both lemon varieties began defoliation from November 11–November 20, with peak defoliation from November 21-December 10, and defoliation ending between December 11-December 20. Five rootstock types were selected for comparison of rootstock combinations with 'Allen Eureka' scions, and the five rootstocks were selected from *C.junos* (Sieb.) Tanaka, *C.grandis* (L.) Osbeck, *C.limonia*, Valencia orange (nondeciduous rootstock in winter), and *Poncirus trifoliate* (L.) Raf (deciduous rootstock in winter). Nine plants were grafted with each rootstock type, and an average plant height of 60 cm was transplanted. A total of 1 year after transplanting, it was found that different rootstocks had a great influence on the growth of 'Allen Eureka'. The 'Allen Eureka'-scion/*C.junos* (Sieb.) Tanaka-rootstock combination had the weakest growth, with an average plant height of 90 cm; the 'Allen Eureka'/Valencia orange had the strongest growth, with an average plant height of 180 cm; and the 'Allen Eureka'-scion/*Poncirus trifoliate* (L.) Raf-rootstock combination had medium growth, with an average plant height of 120 cm. Abnormal winter defoliation was also observed in all five rootstocks grafted to 'Allen

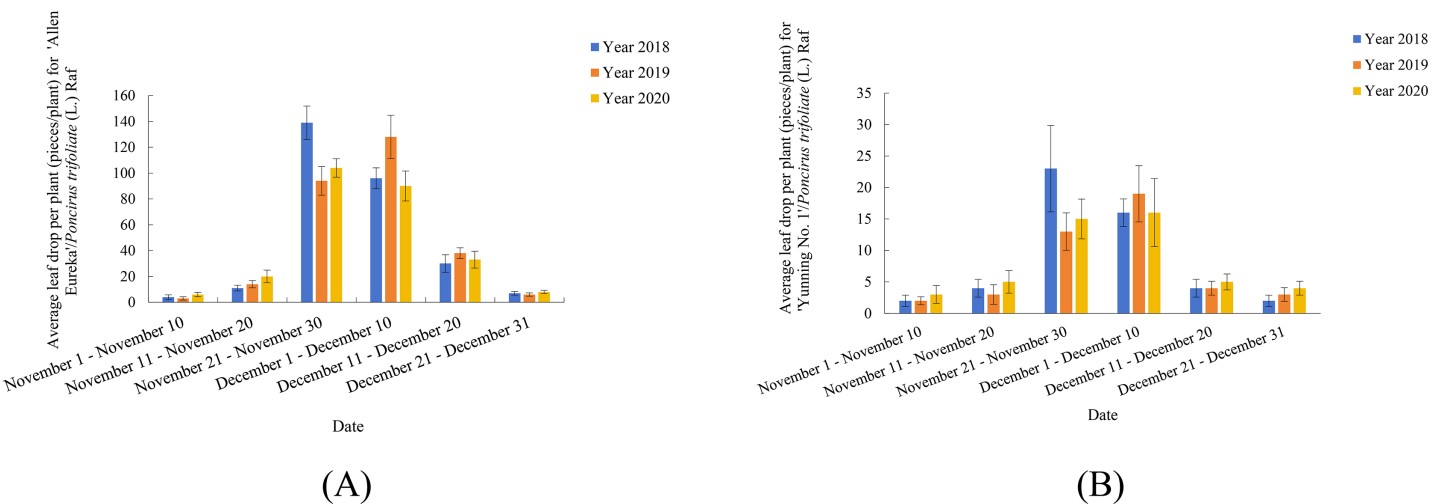

**Figure 1** (A) 'Allen Eureka' November-December Leaffall Statistics 2018–2020. (B) 'Yunning No. 1' November-December Leaffall Statistics 2018–2020.                                                               

Eureka', and the heaviest defoliation was in the 'Allen Eureka'-scion/*Poncirus trifoliate* (L.) Raf-rootstock combination, with cumulative winter defoliation of 180 leaves/plant. The lowest defoliation was in the 'Allen Eureka'-scion/*C.grandis* (L.) Osbeck-rootstock combination with a cumulative winter defoliation of 117 leaves/plant, which indicated that this defoliation state of 'Allen Eureka' was stable and that the defoliation state of 'Allen Eureka' could not be solved by changing the rootstocks. Second, the defoliation of the 'Yunning No. 1'-scion/*Poncirus trifoliate* (L.) Raf-rootstock was also compared with that of the 'Allen Eureka'-scion/*Poncirus trifoliate* (L.) Raf-rootstock, and the cumulative winter defoliation of 'Yunning No. 1'-scion/*Poncirus trifoliate* (L.) Raf-rootstock, at the same time and under the same cultivation conditions, was 23 leaves/plant, which is a normal physiological defoliation of old leaves. The results of the period study showed that 'Allen Eureka' and 'Yunning No. 1' exhibited significant varietal differences.

For experimental materials, the scions of two lemon varieties, 'Allen Eureka' (Allen Eureka) and 'Yun Lemon No. 1' (Yuning No. 1), were selected, and 'Yun Lemon No. 1' of the same defoliation period was used as a control and grafted onto the annual *Poncirus trifoliate* (L.) Raf rootstocks in November 2019. When the grafted seedlings of the two lemon varieties reached an average plant height of 60 cm, the seedlings of the two lemon varieties with good growth conditions and basically the same size were selected for transplanting and cultivation, respectively. The culture container was a plastic bucket (diameter of the mouth of the bucket 60 cm, diameter of the bottom of the bucket 55 cm, height of the bucket 60 cm), and the culture substrate was cocopeat: red loam soil: organic fertilizer = 6:3:1. One hundred and eighty plants of each variety were planted (a total of 360 plants) and placed in the Institute of Tropical Subtropical Cash Crops, Yunnan Academy of Agricultural Sciences in a rain-sheltered arched shed (97° 87′ E, 24° 2′ N). The height of the shed was 3.2 ~ 3.5 m. General management, the average height of the plants was 200 cm and the average crown width was 150 cm, for the experimental sampling.

## Sampling

Based on the observation and recording of the phenological period of two lemon varieties, 'Allen Eureka' and 'Yunning No. 1', sampling was carried out according to the time of defoliation and the amount of defoliation in three periods, namely, the pre-defoliation stage (the beginning of defoliation, CQ), the mid-defoliation stage (a large amount of defoliation, CZ), and the post-defoliation stage (near the end of defoliation, CH). The sampling times were November 15, 2021, December 1, 2021, and December 15, 2021 (Table S1). Each variety was divided into nine groups (three groups for enzyme activity determination, three groups for hormone determination, and three groups for transcriptome determination). First, 1.0 g of the middle and lower mature petioles of barrel-planted plants from different treatment periods were collected from the AZ (0.3–0.5 cm at the base of the petiole) (Fig. S1), and the samples were transferred to liquid nitrogen quickly frozen after sample collection and placed at −80 °C for freezing and preservation for future use.

## Measurement of enzyme activities

A kit was used to determine superoxide dismutase (SOD) activity, peroxidase (POD) activity, catalase (CAT) activity, malondialdehyde (MDA) content, cellulase activity, and pectinase activity, and was biologically repeated three times. The primary test instrument was an MD SpectraMax 190 full-wavelength enzyme labeler.

## Hormone determination

In this experiment, the 1-aminocyclopropane-1-carboxylic acid (ACC), abscisic acid (ABA), auxin (IAA), and gibberellin (GA3) contents of the two lemon varieties at three periods were measured using high-performance liquid chromatography-tandem mass spectrometry (HPLC–MS/MS), repeated three times. The main testing instruments were an Agilent 1290 high-performance liquid chromatograph (Agilent, Santa Clara, CA, USA) and a SCIEX-6500 Qtrap (MSMS) (AB SCIEX, Framingham, MA, USA).

## RNA extraction, RNA-seq library preparation, and transcriptome sequencing

Total RNA was extracted from 18 lemon samples using a plant RNA extraction kit (QIAGEN, Hilden, Germany), and three periods of two lemon varieties, three biological replicates (named ACQ-1, ACQ-2, ACQ-3, ACZ-1, ACZ-2, ACZ-3, ACH-1, ACH-2, ACH-3, YCQ-1, YCQ- 2, YCQ-3, YCZ-1, YCZ-2, YCZ-3, YCH-1, YCH-2, and YCH-3), and three technical replicates were performed for each sample. After that, mRNA was purified using magnetic beads and Oligo (dT), and the mRNA was broken up into small pieces by adding fragmentation buffer. Using six-base random hexamers as a template and mRNA as the starting material, the primary cDNA strand was created. The second cDNA strand was subsequently produced by adding buffer, dNTPs, RNase H, and DNA polymerase I. After purification with the QiaQuick PCR kit and elution with EB buffer, the second cDNA strand was end-repaired, poly (A) was added, the sequencing junctions were connected, the fragment sizes were selected by agarose gel electrophoresis, and then PCR

was performed. The sequencing library was constructed and then sequenced with Illumina HiSeq 6000.

## Transcriptome data analysis

In order to remove junctions and low-quality data from the transcriptome data, the raw data were filtered using fastp software (https://github.com/OpenGene/fastp) to remove reads containing junctions, reads with an N-ratio of 10% or more, and reads with a low-quality (quality value less than 20) base ratio of more than 50%. Finally, FastQC software was used to check the quality of the clean data, and subsequent analysis was carried out after passing the quality control. All downstream analyses were based on sequence data. We downloaded lemon reference genome and gene model annotation files directly from the Genome website (https://www.citrusgenomedb.org/jbrowse/index.html?data=data/Climon_Alt_v1.0f) (*Bao et al., 2023*). The index of the reference genome was built using HISAT2 v2.0.5, and paired-end clean reads were aligned to the reference genome using HISAT2 v2.0.5 (*Kim, Langmead & Salzberg, 2015*; *Shirasawa et al., 2017*). The mapped reads of each sample were assembled by StringTie (v1.3.3b) (*Pertea et al., 2015*) in a reference-based approach. With $|\log_2\text{Fold Change}| \geq 1$ and $p$adj $\leq 0.05$ as the thresholds, differential expression analysis of two conditions/groups (two biological replicates per condition) was performed using the DESeq2 R package (1.20.0). Gene Ontology (GO) and KEGG pathway enrichment analysis of differentially expressed genes (DEGs) was implemented using the clusterProfiler R package.

## RT-qPCR analysis

The expression levels of five DEGs selected from the RNA-seq data were verified by RT-qPCR using the cDNA samples used for RNA-Seq library construction. Primer 5.0 software was used to design primer pairs. The primer sequence was synthesized by Beijing Tian Yi Hui Yuan Biotechnology Co., Ltd. (Beijing, China), and the primer information is shown in Table S2. RT-qPCR was performed using a Roche LightCycler 480 with the following procedure: 95 °C for 5 min, followed by 40 cycles at 95 °C for 10 s, 60 °C for 10 s, and 60°C for 30 s. The housekeeping gene Actin was used as an internal control. The relative expression level was calculated by the $2^{-\Delta\Delta Ct}$ method.

## Statistics and analysis of experimental data

Gene expression data were expressed as the mean ± standard deviation. The significance between means was statistically analyzed using a *t*-test ($p < 0.05$). Microsoft Office Excel 2016 was utilized for statistical processing of data as well as graphing, and the results were expressed as the mean ± standard deviation. Gene coexpression networks were constructed using the R software WGCNAv1.69 package.

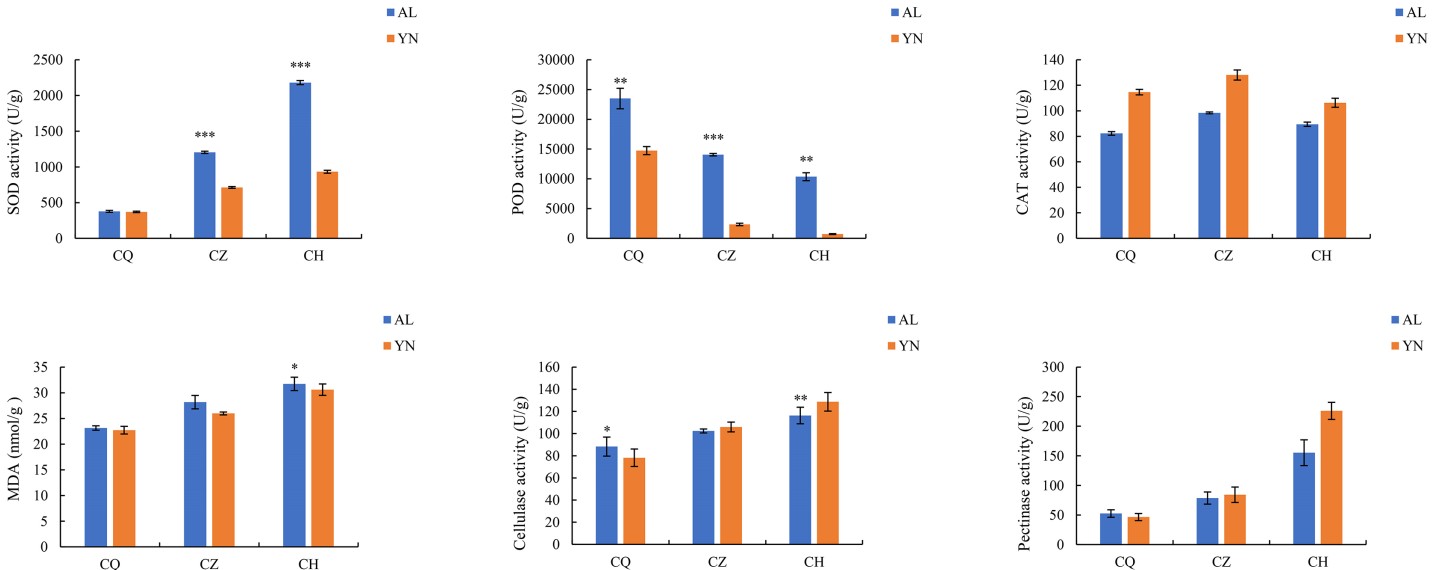

**Figure 2 Changes in enzyme activities during different defoliation periods in two varieties, 'Allen Eureka' and 'Yunning No. 1'.** Significance is the difference between the two varieties; $^*p < 0.05$, $^{**}p < 0.01$, $^{***}p < 0.001$.           

## RESULTS

### Changes in enzyme activities during different defoliation periods in two lemon varieties

The SOD, MDA, cellulase, and pectinase activities of the two lemon varieties, 'Allen Eureka' and 'Yunning No. 1', all gradually increased with prolonged defoliation time. The content of SOD in 'Allen Eureka' was significantly higher than that of 'Yunning No. 1' in the mid- and post-defoliation stages of defoliation. CAT activity tended to increase and decrease with prolonged defoliation, but the magnitude of change was small in both varieties. POD activities all showed a significant decreasing trend, with the prolongation of defoliation time being significantly lower in the mid-defoliation stage (CZ) and post-defoliation stage (CH) than in the pre-defoliation stage (CQ), and the differences between the two varieties were distinct (Fig. 2). The protective enzymes SOD, POD, and CAT showed an upward trend in a short period of time when the plants faced adversity, but as the stress intensified or the stress time prolonged, the activities of the protective enzymes eventually declined, which may be attributed to the fact that the duration and intensity of the stress exceeded the upper limit of the plant's tolerance. Although both SOD and POD were effective in removing excess reactive oxygen species and superoxide anion radicals from the cells, the MDA content of both lemon varieties kept increasing, indicating that the activities of the two enzymes were not sufficient to remove reactive oxygen species and superoxide radicals produced in the cells throughout the defoliation process. Increased cellulase and pectinase activities can both lead to faster cell wall degradation and accelerated leaf abscission.

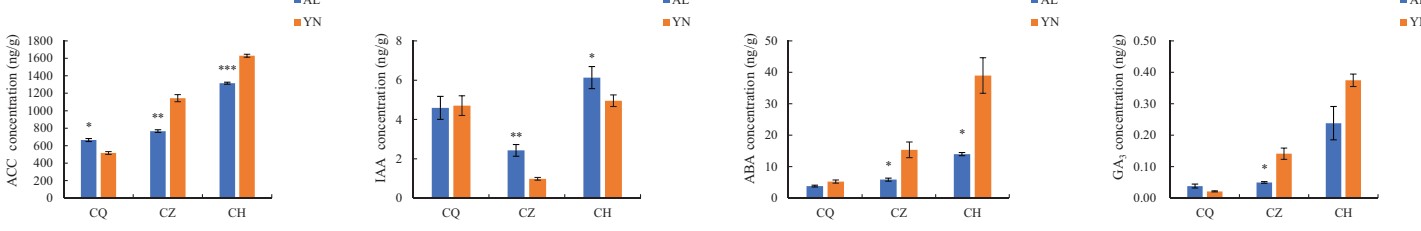

**Figure 3 Changes in endogenous hormone contents during different defoliation periods in two lemon varieties, 'Allen Eureka' and 'Yunning No. 1'.** Significance is the difference between the two varieties; $*p < 0.05$, $**p < 0.01$, $***p < 0.001$.

## Changes in endogenous hormone contents during different defoliation stages in two lemon varieties

The ACC, ABA, and GA3 content of the two lemon varieties, 'Allen Eureka' and 'Yunning No. 1', gradually increased with the prolongation of defoliation time, and the ACC, ABA, and GA3 content of 'Yunning No. 1' was significantly higher than that of 'Allen Eureka' in both the mid- and post-defoliation stages. The IAA content of the two lemon varieties showed a decreasing trend from the pre-defoliation stage to the mid-defoliation stage. At the mid-defoliation stage, the IAA content of 'Allen Eureka' was significantly higher than that of 'Yunning No. 1', and the IAA content of the two lemon varieties increased again from the mid-defoliation stage to the post-defoliation stage (Fig. 3). Increased levels of ACC and ABA promote organ abscission. IAA both inhibits and promotes abscission, and its effect on organ abscission is related to the concentration, timing, and application site of the growth hormone used. The role played by GA in plant organ abscission is still controversial.

## Statistics of off-region transcriptome sequencing data

After sequencing quality analysis, 18 libraries were constructed from three replicates across three periods of 'Allen Eureka' and 'Yunning No. 1'. A total of 116.35 Gb of clean data was obtained for 18 samples with Q30 > 89.15%. The number of raw reads ranged from 39,849,322 to 46,685,856. Both clean and uniquely mapped reads made up more than 70% (Table 1; Table S3), indicating that the reference genome was selected appropriately. The Pearson correlation coefficients between the biological replicates for each sample ranged from 0.89 to 0.97. The 18 samples were subjected to principal component analysis (PCA). After dimensionality reduction, it was found that the samples of the same variety and period were clustered together (Fig. 4), and the results of the analysis showed good reproducibility of all the samples used for this experiment.

## Number of DEGs at different defoliation stages in two lemon varieties

Two-by-two comparisons of the 'Allen Eureka' and 'Yunning No. 1' genotypes were performed at the pre-defoliation, mid-defoliation, and post-defoliation stages of the two lemon varieties to identify lemon defoliation-related genes. When compared to 'Yunning No. 1', 898 genes in 'Allen Eureka' were significantly differentially expressed, 320 genes were significantly upregulated, and 578 genes were significantly downregulated in the pre-defoliation stage (CQ). At the mid-defoliation stage (CZ), 4,856 genes were
**Table 1 Sample sequencing and data comparison statistics.**

| Sample | Clean reads | Q20(%) | Q30(%) | GC content(%) | Total_map | Unique_map |
|--------|-------------|--------|--------|---------------|-----------|------------|
| ACQ_1 | 45,389,670 | 96.02 | 89.77 | 43.37 | 36,144,885 (83.66%) | 33,679,703 (77.96%) |
| ACQ_2 | 43,962,366 | 97.36 | 92.64 | 43.12 | 33,450,174 (80.81%) | 31,320,726 (75.66%) |
| ACQ_3 | 44,361,426 | 97.58 | 93.14 | 42.25 | 35,765,162 (83.80%) | 33,420,700 (78.30%) |
| ACZ_1 | 45,955,674 | 97.52 | 93.00 | 43.49 | 36,989,230 (86.88%) | 34,404,354 (80.81%) |
| ACZ_2 | 43,898,930 | 97.73 | 93.39 | 43.50 | 35,754,881 (87.02%) | 33,247,982 (80.92%) |
| ACZ_3 | 45,156,658 | 97.56 | 93.03 | 43.41 | 37,323,752 (87.19%) | 34,705,696 (81.08%) |
| ACH_1 | 43,731,580 | 95.77 | 89.15 | 43.09 | 36,219,818 (84.73%) | 33,715,982 (78.87%) |
| ACH_2 | 44,868,396 | 97.62 | 93.19 | 42.79 | 36,712,274 (85.54%) | 34,109,504 (79.47%) |
| ACH_3 | 46,175,892 | 97.44 | 92.74 | 42.93 | 38,059,126 (85.34%) | 35,362,964 (79.30%) |
| YCQ_1 | 47,139,164 | 96.11 | 89.81 | 43.31 | 39,595,137 (86.20%) | 36,655,291 (79.80%) |
| YCQ_2 | 46,281,504 | 97.56 | 93.07 | 42.39 | 37,662,531 (86.22%) | 34,931,334 (79.97%) |
| YCQ_3 | 44,404,670 | 97.52 | 92.96 | 42.48 | 36,165,422 (85.88%) | 33,663,423 (79.94%) |
| YCZ_1 | 47,499,598 | 96.34 | 90.31 | 43.99 | 36,921,295 (79.08%) | 34,468,315 (73.83%) |
| YCZ_2 | 43,595,010 | 97.55 | 93.01 | 43.12 | 33,412,437 (80.05%) | 31,140,010 (74.61%) |
| YCZ_3 | 45,732,154 | 96.85 | 91.47 | 44.05 | 36,180,101 (82.22%) | 33,649,047 (76.46%) |
| YCH_1 | 46,918,934 | 96.13 | 89.93 | 43.54 | 38,598,212 (84.09%) | 35,842,756 (78.09%) |
| YCH_2 | 44,498,312 | 97.65 | 93.25 | 43.26 | 35,834,006 (84.63%) | 33,173,821 (78.35%) |
| YCH_3 | 41,927,840 | 97.79 | 93.70 | 44.95 | 31,449,796 (78.92%) | 29,098,680 (73.02%) |

significantly differentially expressed, 2,513 genes were significantly upregulated, and 2,343 genes were significantly downregulated. At the post-defoliation stage (CH), 3,126 genes were significantly differentially expressed, with 1,336 genes significantly upregulated and 1,790 genes significantly downregulated (Fig. 5A), and the most DEGs in the mid-defoliation stage (CZ) out of the three periods (Table S4). A Venn diagram (Fig. 5B) was plotted based on all the DEGs across the three periods, and a total of 143 DEGs in the three periods compared with 400 (CQ), 3,621 (CZ), and 1,990 (CH) genes were expressed in only one period.

## GO enrichment analysis of DEGs at different defoliation stages in two lemon varieties

Additionally, GO enrichment analysis showed that 99 (pre-defoliation stage CQ), 338 (mid-defoliation stage CZ), and 378 (post-defoliation stage CH) DEGs were annotated to the GO database for the 'Allen Eureka' and 'Yunning No. 1' lemons in the three periods, respectively. According to the significance level of GO term enrichment, the differential genes of the two varieties were mainly enriched in the molecular functional classifications of xyloglucosyl transferase activity, glucosyltransferase activity, transferring glycosyl groups, iron ion binding, oxidoreductase activity, hydrolase activity, DNA-binding transcription factor activity, transcriptional regulatory activity, and classification of cellular components of the apoplast, extracellular region, cell wall, external encapsulating structure, and cell periphery. In addition, biological processes included cellular glucan
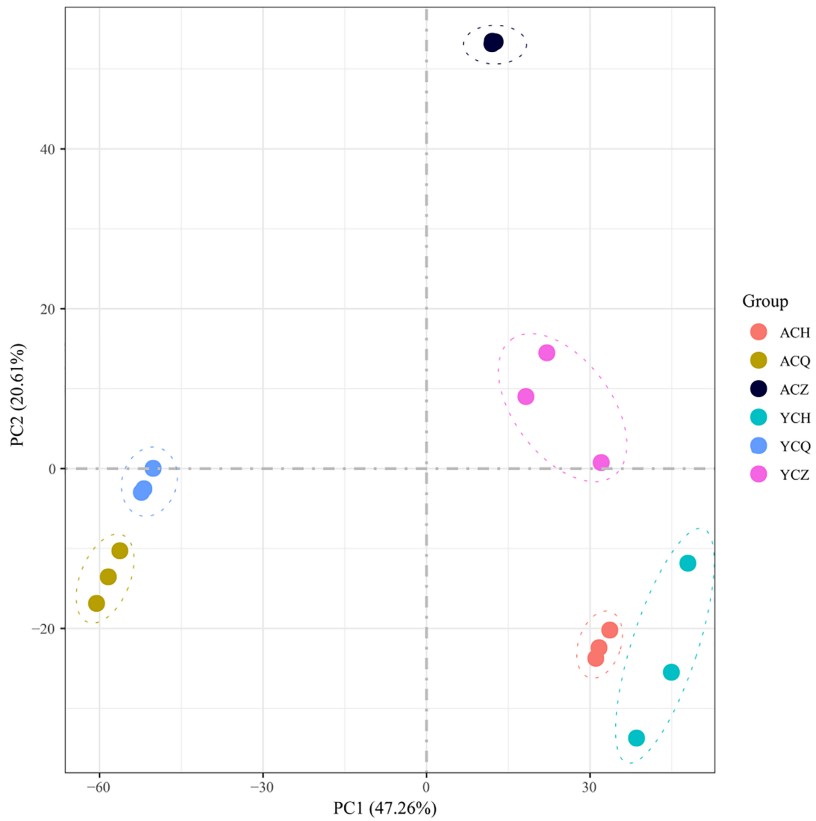

**Figure 4 Principal component analysis (PCA) of differentially expressed genes in 'Allen Eureka' and 'Yunning No. 1' lemons at each defoliation period.**

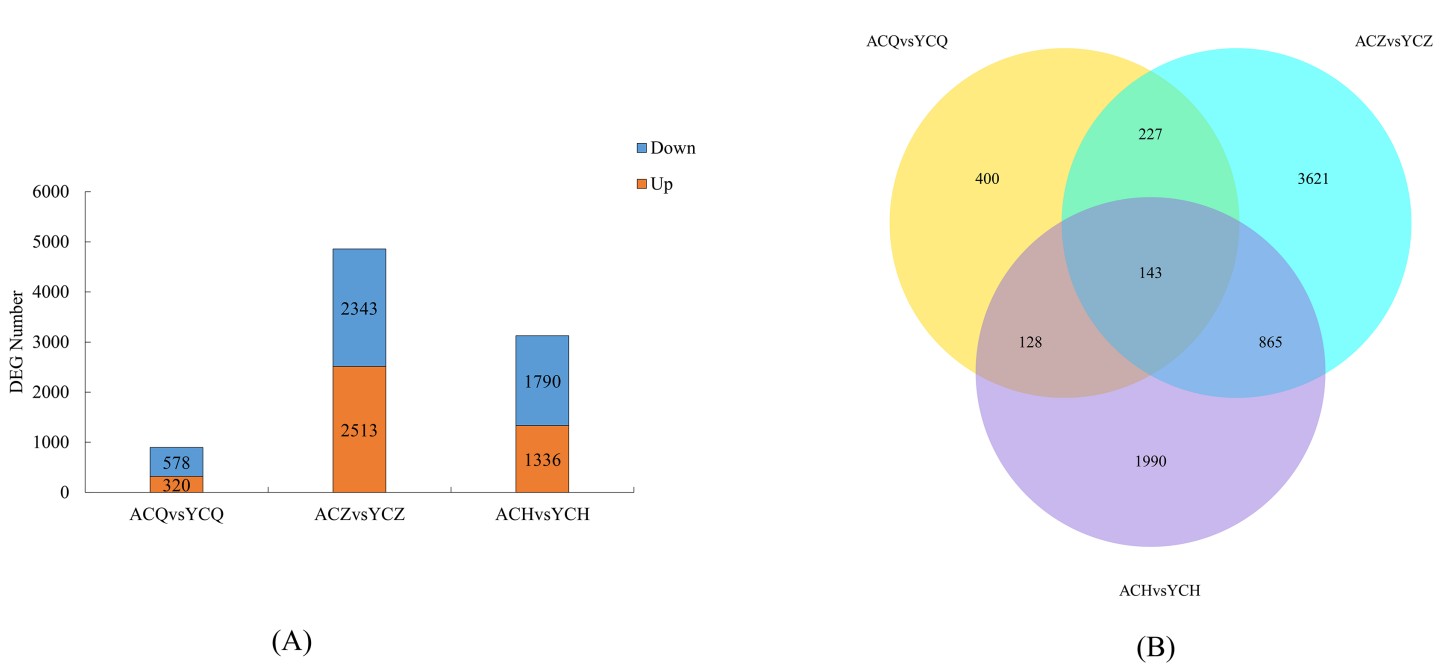

**Figure 5 (A) Number of DEGs between 'AL' and 'YN' for each defoliation period. DEGs were screened according to |log₂ (fold change) | ≥ 1 and padj ≤ 0.05. (B) Venn diagram of 'Allen Eureka' and 'Yunning No. 1' for each defoliation period.**

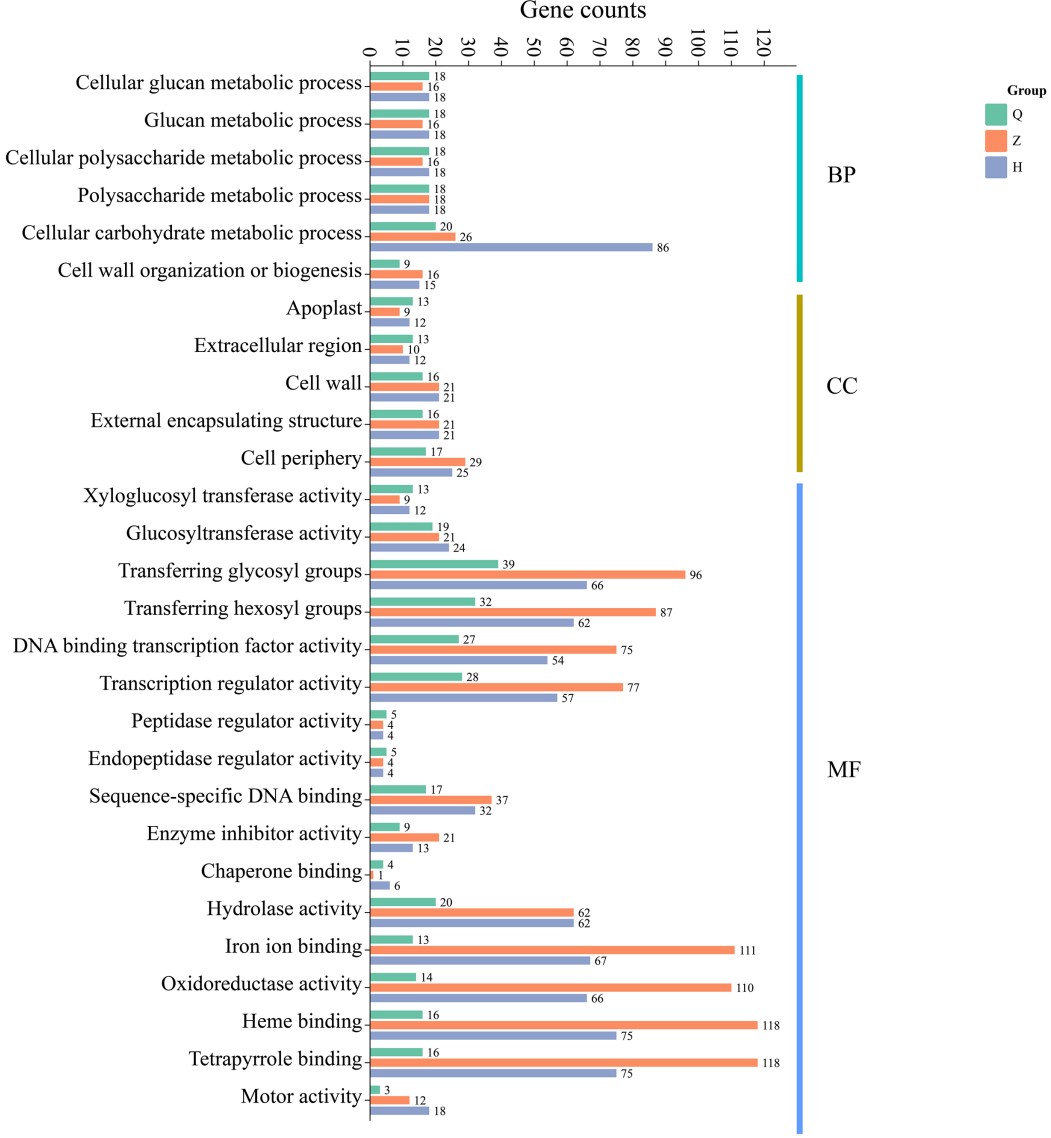

**Figure 6 GO functional enrichment of differentially expressed genes in 'Allen Eureka' and 'Yunning No. 1' lemons at each defoliation period.**

metabolic, glucan metabolic, cellular polysaccharide metabolic, polysaccharide metabolic, and cellular carbohydrate metabolic processes (Fig. 6). The related differential genes in molecular functions, cellular components, and biological process classifications were significantly different in all three periods, with most of them enriched to significantly higher differential genes in the mid-defoliation stage (CZ) GO than in the pre-defoliation stage (CQ) and post-defoliation stage (CH).

## KEGG enrichment analysis

The results of KEGG metabolic pathway enrichment of DEGs between two lemon varieties, 'Allen Eureka' and 'Yunning No. 1', at three periods (Fig. 7) showed that the number of DEGs and related metabolic pathways enriched at the mid-defoliation stage

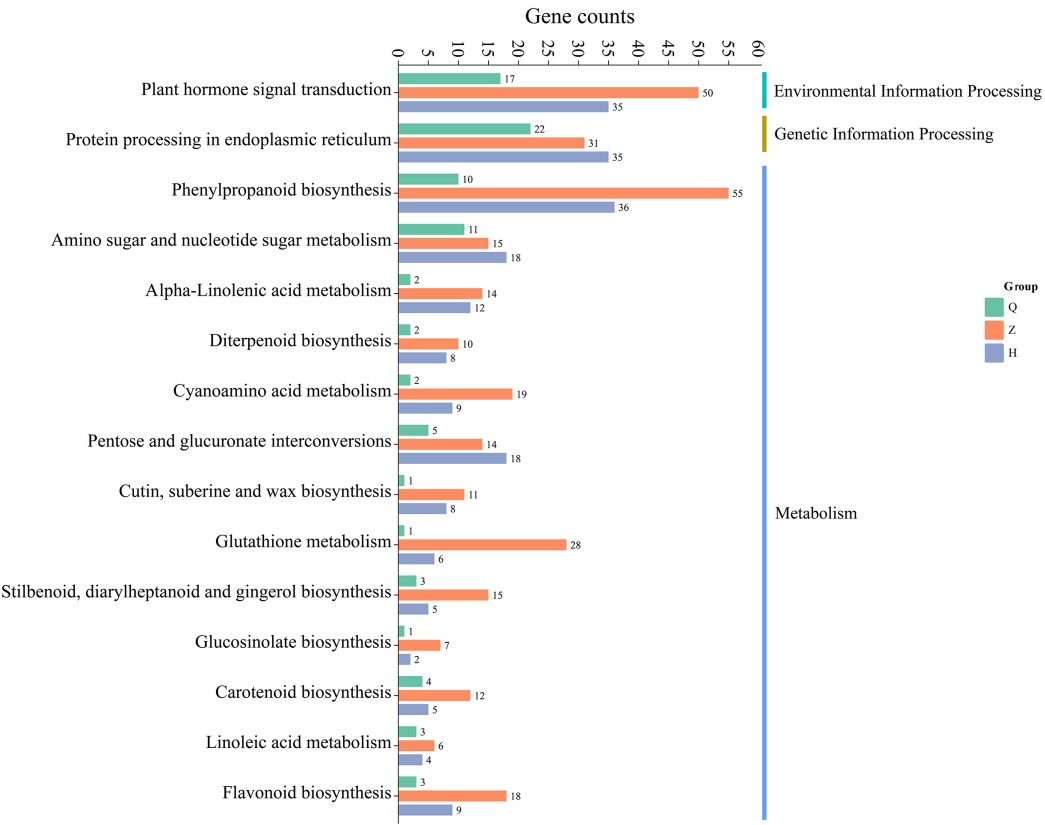

**Figure 7 Enrichment of KEGG metabolic pathways of differentially expressed genes in 'Allen Eureka' and 'Yunning No. 1' lemons at each defoliation period.**

(CZ) was the highest, which mainly focused on phenylpropanoid biosynthesis, plant hormone signal transduction, protein processing in the endoplasmic reticulum, amino sugar and nucleotide sugar metabolism, and glutathione metabolism, cyanoamino acid metabolism. The number of differential genes and related metabolic pathways enriched in the post-defoliation stage (CH) was high, which focused on phenylpropanoid biosynthesis, plant hormone signal transduction, protein processing in the endoplasmic reticulum, amino sugar and nucleotide sugar metabolism, pentose and glucuronate interconversions, and alpha-linolenic acid metabolism pathways. Relatively few metabolic pathways were enriched in the pre-defoliation stage (CQ), which mainly focused on protein processing in the endoplasmic reticulum, plant hormone signal transduction, amino sugar and nucleotide sugar metabolism, and phenylpropanoid biosynthesis pathways.

### DEGs of the plant hormone signal transduction pathway

The KEGG enrichment analysis revealed that the DEGs between 'Allen Eureka' and 'Yunning No. 1' lemons were more abundant in all three periods of the plant hormone signal transduction pathway, and comparing the three periods, the mid-defoliation stage (CZ) had the most DEGs in the plant hormone signal transduction pathway. Among them, *Aux/IAA* (*CL9G066930012_alt, CL3G046631012_alt*, and *CL3G046634012_alt*), *SAUR*

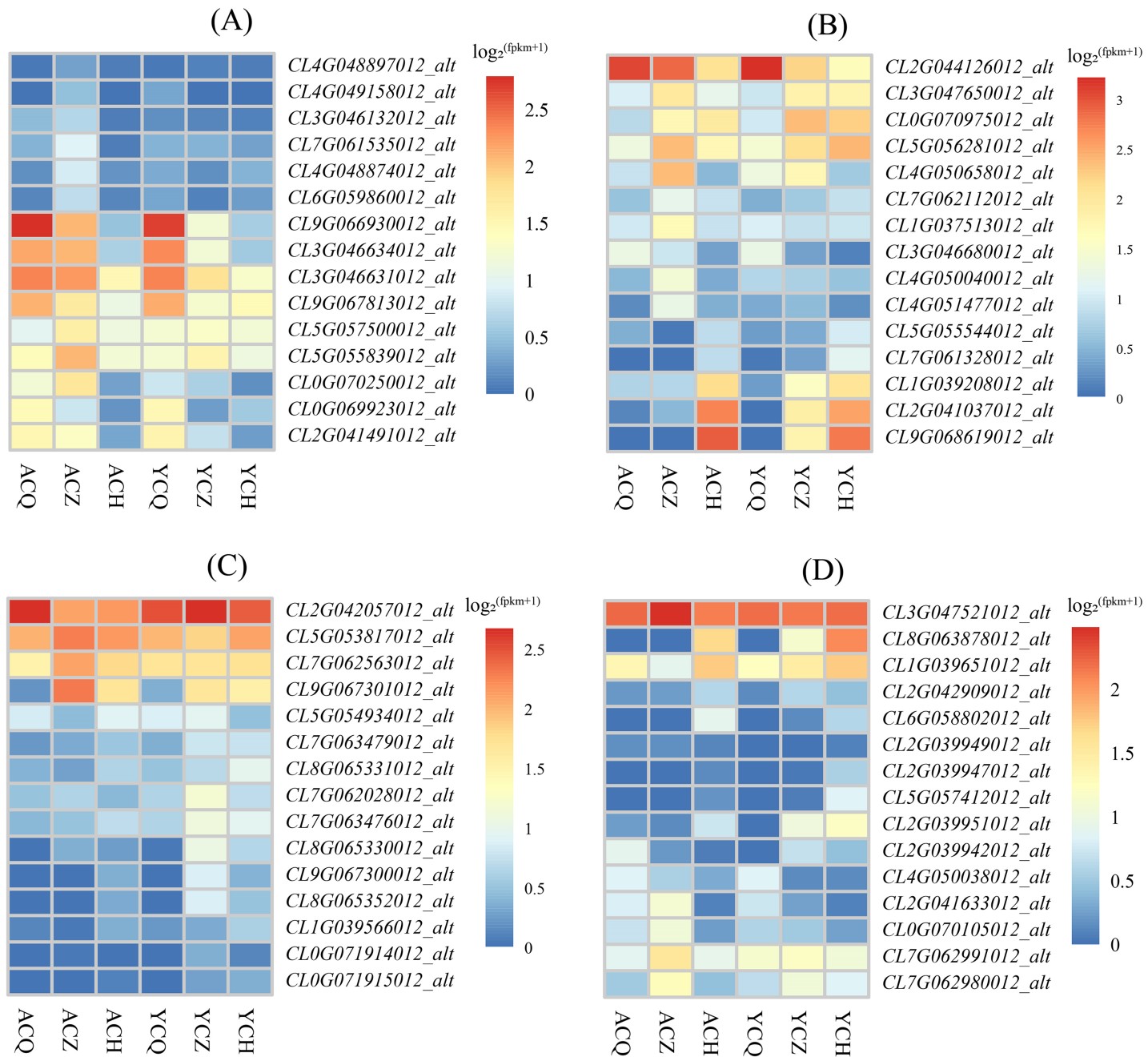

**Figure 8 Expression characteristics of differentially expressed genes of metabolic pathways in each de-foliation period of the 'Allen Eureka' and 'Yunning No. 1' lemons.** (A) Plant hormone signal transduction; (B) phenylpropanoid biosynthesis; (C) glutathione metabolism; (D) alpha-linolenic acid metabolism. The horizontal coordinates of the graph in the heatmap are the sample names, and the vertical coordinates are the values of the differentially expressed genes after normalization of FPKM; the redder the color, the higher the expression, and the bluer the expression, the lower the expression.               

(*CL0G069923012_alt*), GH3 (*CL2G041491012_alt*), and the two-component response regulator *ARR* (*CL9G067813012_alt*) were downregulated in the mid-defoliation stage compared to the pre-defoliation stage, and the relative expression of 'Allen Eureka' was significantly higher than that of 'Yunning No. 1' (Fig. 8A). The expression of growth

hormone-responsive genes Aux/IAA, SAUR, and GH3-related genes were all down-regulated, suggesting that the IAA signaling pathway was inhibited, and that the massive leaf abscission in the middle stage of defoliation might be related to the down-regulation of these growth hormone-responsive genes.

### Metabolic pathway DEGs

As a result of KEGG enrichment analysis, the DEGs between the 'Allen Eureka' and 'Yunning No. 1' lemons were also relatively concentrated in metabolic pathways, especially phenylpropanoid biosynthesis (Fig. 8B), glutathione metabolism (Fig. 8C), and the alpha-linolenic acid metabolism pathway (Fig. 8D). The phenylpropane biosynthesis pathway was characterized by β-glucosidase (*CL4G050040012_alt, CL7G062112012_alt*), peroxidase (*CL3G047650012_alt, CL0G070975012_alt, CL5G056281012_alt, CL2G041037012_ alt, CL9G068619012_alt*), and transferase (*CL1G037513012_alt, CL4G051477012_alt*) expression differences, significantly in all three defoliation periods. Among them, the expression of β-glucosidase was significantly higher in both 'Allen Eureka' and 'Yunning No. 1' at the mid-defoliation stage. β-glucosidase has a role in degrading cell walls to promote organ abscission. Peroxidase was significantly higher in both the mid- and post-defoliation stages than in the pre-defoliation stage, and the expression of 'Allen Eureka' was significantly higher than that of 'Yunning No. 1' in both the mid-defoliation stages *CL4G050040012_alt* and *CL7G062112012_alt*. The regulatory mechanism of peroxidase on plant organ abscission is mainly realized through the participation in the oxidation process of growth hormone, and the enzyme is able to reduce the level of AZ growth hormone and promote the abscission of plant organs.
The expression of transferases (*CL1G037513012_alt, CL4G051477012_alt*) was also significantly higher in 'Allen Eureka' than in 'Yunning No. 1' at the mid-defoliation stage. The expression of glutathione S-transferases (*CL5G053817012_alt, CL9G067301012_alt*, and *CL7G062563012_alt*) in the glutathione metabolism pathway was significantly higher in the mid-defoliation stage of 'Allen Eureka' than in 'Yunning No. 1'. The expression of lipoxygenase (*CL7G062991012_alt, CL7G062980012_alt*) and cytochrome P450 (*CL0G070105012_alt, CL2G041633012_alt*) in the α-linolenic acid metabolism pathway was significantly higher in 'Allen Eureka' than in 'Yunning No. 1' at the mid-defoliation stage. Lipoxygenase affects the oxidation of plant fats, which in turn affects plant senescence, death, and abscission. Enhanced lipoxygenase activity may have induced the abscission of lemon leaves. All of the above genes were expressed in 'Allen Eureka' at higher levels than in 'Yunning No. 1', which may have contributed to the fact that 'Allen Eureka' was more prone to defoliation.

## Construction of gene coexpression networks

To gain insight into the coexpressed genes of the two lemon varieties at three defoliation stages (CQ, CZ, and CH) and to mine genes related to defoliation height, weighted gene co-expression network analysis (WGCNA) was performed. By screening the weight values, β = 14 was finally chosen to construct the network, the dynamic shear tree method was used to merge the modules with similar expressions, and a total of 11 coexpression

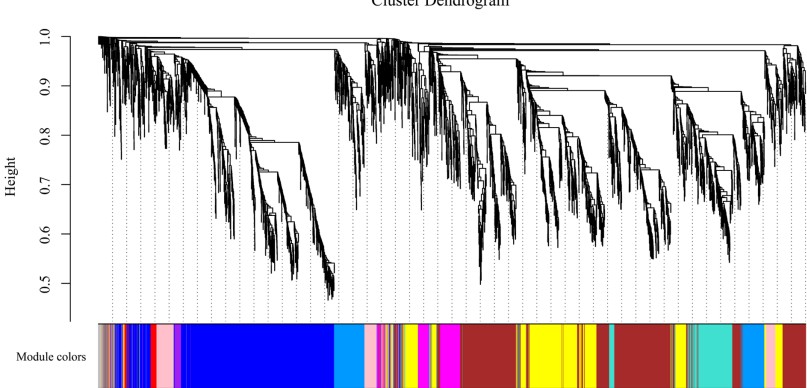

**Figure 9 Gene clustering tree and module cuts.** Merging gene modules with similar expression patterns; different colors represent different modules, the number of genes assigned in a module is clustered ac-cording to their expression in a correlation, and genes with higher clusters are assigned to a module.

modules were obtained (Fig. 9). The turquoise module contained the highest number of genes (2,926 genes), followed by the blue module containing 1,923 genes, and the lowest number of genes was within the gray module, which contained 69 genes. The average number of genes contained in each module was 737.

Four of the 11 modules were highly specific to the sample. One was present in each of the pre-mid-defoliation stages, and two were present in the post-defoliation stage (Fig. S2). Five reported defoliation-associated genes (*CL2G044356012_alt, CL6G058882012_alt, CL9G068078012_alt, CL1G038862012_alt*, and *CL3G046093012_alt*) were found in the blue module by enrichment analysis. Therefore, the present study focused attention on the blue module, screened the genes with high connectivity in the module, and visualized the network using Cytoscape software (Fig. S3). Six genes of ribosomal proteins (*CL6G060070012_alt, CL1G038121012_alt, CL6G059277012_alt, CL7G060527012_alt*, and *CL9G068716012_alt*) and gibberellin-regulated proteins (*CL5G054544012_alt*) were found to have the connectivity of the module ranked in the top 1% of the blue module, which can be used as the hub genes in the module. In addition, *WRKY* genes (*CL6G058882012_alt* and *CL1G038862012_alt*) ranked low in connectivity in the module, but these two genes were presumed to be key genes in the leaf abscission regulatory pathway. Organ shedding involves two important processes, cell division and hydrolase induction, both of which are based on active metabolism. The protein and RNA content of the AZ significantly increase during abscission, and various metabolic inhibitors and protein synthesis inhibitors inhibit the formation of the AZ. Gibberellin regulates organ shedding and has a range of effects that are more pronounced at higher concentrations. WRKY transcription factors are mainly expressed during organ abscission in response to abscisic acid and ethylene, which in turn regulate organ abscission.

## RT-qPCR validation of DEGs

In this study, we also used RT-qPCR to validate five genes involved in plant hormone synthesis, signaling, and other processes related to plant organ abscission: peroxidase

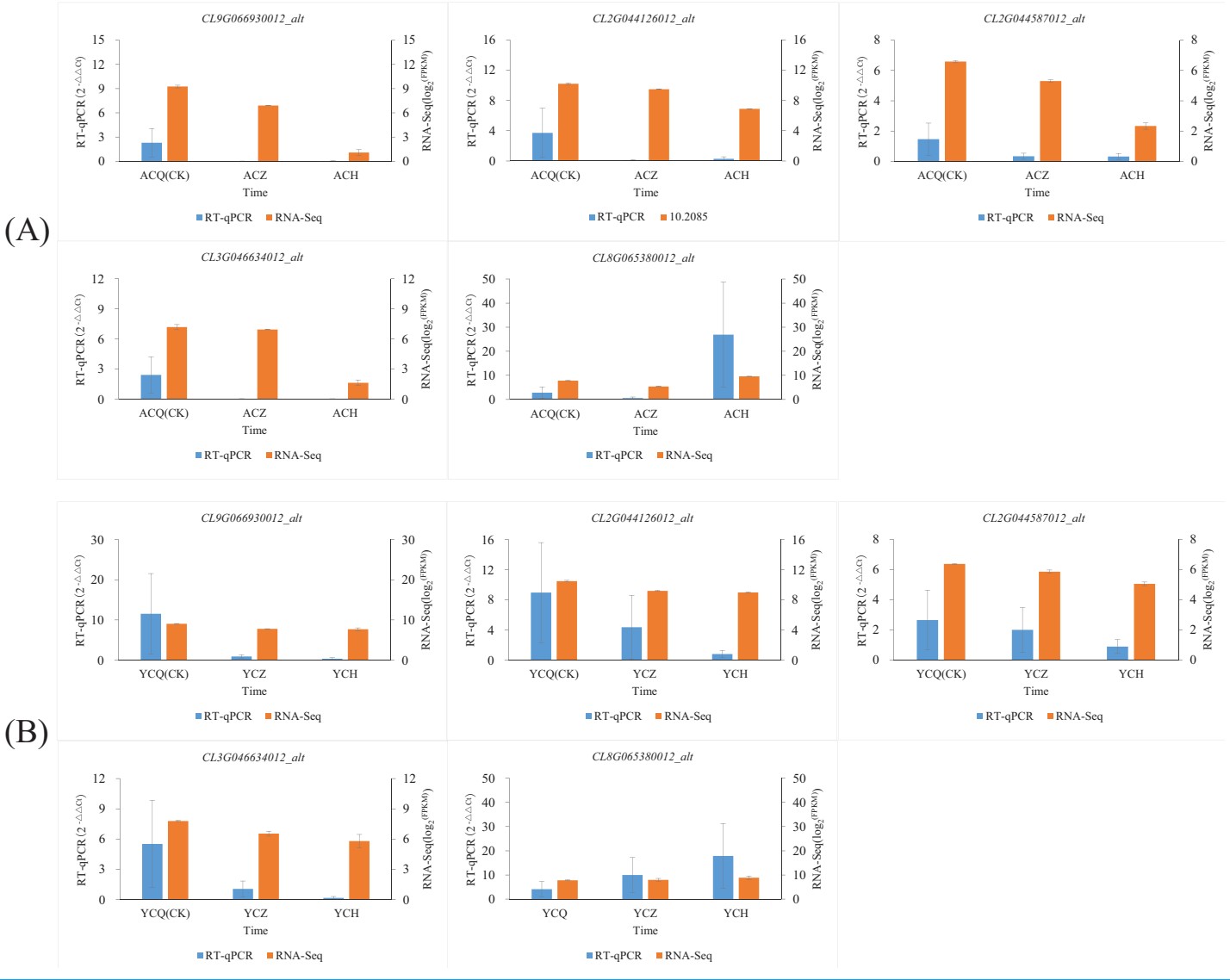

**Figure 10 Validation of the expression levels of five differentially expressed genes.** (A) Validation of expression levels of five differentially expressed genes in 'AL'. (B) Validation of expression levels of five differentially expressed genes in 'YN'. To observe more clearly the trend of changes in the expression of individual genes after ethylene glycol treatment, we normalized the expression of genes using $\log_2$ (FPKM).

(*CL2G044126012_alt*), Aux/IAA gene family (*CL3G046634012_alt, CL9G066930012_alt*), weak ethylene-insensitive protein (*CL2G044587012_alt*), and protein phosphatase (*CL8G065380012_alt*). The results (Fig. 10; Table S5) showed that the changes in the fluorescence quantitative expression levels of the five DEGs converged with the changes in transcriptome gene abundance.

## DISCUSSION

The abscission signal leading to the process of organ abscission is mainly a number of cell wall degrading enzymes and proteins in action, of which peroxidase, pectinase, and cellulase are some of the main functional enzymes of plant organ abscission. In this study,

we found that SOD, cellulase, pectinase, and MDA activities of two lemon varieties, 'Allen Eureka' and 'Yunning No. 1', showed a gradual increase with prolonged defoliation time. This suggests that the accumulation of SOD, hydrolytic enzymes, and MDA can promote lemon leaf abscission. Increased activity of hydrolytic enzymes (cellulase and pectinase) can accelerate cell wall degradation, leading to leaf abscission, and the β-glucosidase in cellulose is closely correlated with abscission (*Goldental-Cohen et al., 2017*). In this study, we found that *β-glucosidase* (*CL4G050040012_alt, CL7G062112012_alt*) in the phenylpropane biosynthesis pathway, was both significantly higher in the mid-defoliation stage compared to the pre-defoliation stage, and the expression of 'Allen Eureka' was also significantly higher than that of 'Yunning No. 1' in the mid-foliation stage, suggesting that β-glucosidase also has a facilitating effect on lemon leaf abscission.

In this study, the physiological and biochemical indexes were determined, and it was concluded that the POD activity of both lemon varieties showed a significant decreasing trend with the prolongation of defoliation time. However, transcriptome sequencing results showed that the peroxidase genes (*CL3G047650012_alt, CL0G070975012_alt, CL5G056281012_alt, CL2G041037012_alt,* and *CL9G068619012_alt*) were significantly higher expressed in both the mid- and post-defoliation stages of phenylpropane biosynthesis pathway than in the pre-defoliation stage. Exactly what causes the inconsistency between the physiological and biochemical measurements of POD and the transcriptome sequencing results should be further investigated. It has been shown that peroxidase can catabolize indoleacetic acid (*Intapruk et al., 1994*), reducing the amount of indoleacetic acid in the AZ of the plant and increasing the sensitivity of the AZ to ethylene. Thus, the regulation of plant organ abscission by peroxidase occurs primarily through participation in the oxidative processes of growth factors. In the present study, IAA content was also significantly lower in both lemon varieties at mid-fall compared to pre-fall as determined by hormone assays. This may be due to the catabolism of indoleacetic acid by these peroxidases.

When comparing the plant hormone signal transduction pathways with the highest number of DEGs in the transcriptome sequencing in this article, the relative expression of the Aux/IAA gene family (*CL9G066930012_alt, CL3G046631012_alt,* and *CL3G046634012_alt*), *SAUR* (*CL0G069923012_alt*), and *GH3* (*CL2G041491012_alt*) of the two lemon varieties was significantly down-regulated in both the mid-defoliation stage and post-defoliation stage compared with the pre-defoliation stages. The contents of ACC, ABA, and GA3 showed a gradual increase in both lemon varieties during defoliation as determined by hormone measurements. The combined results of physiological and biochemical indexes and transcriptome sequencing analysis concluded that lemon leaf abscission was negatively regulated by IAA and positively regulated by ACC, ABA, and GA3. This is basically consistent with the results of previous studies. In rose, growth hormone treatment delays rose petal abscission, while silencing of *R. ARF7* promotes petal abscission (*Liang et al., 2020*). Transcriptome sequencing of the abscission region of fallen rose flowers revealed that the AUX/IAA gene family may be involved in this process of abscission (*Jiang et al., 2023*), and that petal abscission can be facilitated by virus-induced gene silencing of *RhIAA16* (*Gao et al., 2016*). Transcriptome analysis of lychee after

ring-stripping defoliation treatment showed that the expression of LcAUX/IAA1 increased in the fruitlet AZ, suggesting that it may have an important role in the abscission process (*Kuang et al., 2012*). *Brassica oleracea* BoGH3.13-1 is expressed in chorioallantoic cells, pollen, the leaf primordium, and in the off-zone of flower abscission, suggesting its possible involvement in these physiological processes (*Jeong et al., 2021*). There are 29 genes related to hormones such as growth hormone, abscisic acid, and ethylene that are differentially expressed in the abscission microsomes of citrus in the off-region transcriptome, including a family of genes related to growth hormone response, including GH3 (*Xie et al., 2018*). LcGH3.1 has increased expression levels in the AZ of small lychee fruits, which may be involved in the regulation of lychee fruit abscission (*Kuang et al., 2012*). Abscisic acid appears to have an abscission-accelerating effect by increasing ACC levels (*Gómez-Cadenas et al., 2000*; *Li et al., 2022*; *Luo et al., 2014*). In a few cases, fruit abscission is determined by the relative concentrations of growth factors and abscisic acid (*Racsko et al., 2006*). Some studies also significantly inhibited fruit abscission by spraying gibberellin GA3 onto *Citrus sinensis* Osbeck (*Chen, Dekkers & Chung, 2006*). However, on other citrus, it has been shown that exogenous gibberellins have no significant difference on shedding (*Gómez-Cadenas et al., 2000*). In a study on the mechanism of abscission in grapevine (*Vitis vinifera* L.), abscission of the floral organs was induced by both exogenous spraying of gibberellins and shading (*Domingos et al., 2015*). Thus, there is controversy regarding the regulation of plant organ abscission by gibberellins.

In this study, WGCNA analysis was performed on two lemon varieties at three defoliation periods, and *WRKY* (*CL6G058882012_alt* and *CL1G038862012_alt*) was found to be significantly up-regulated in the blue module at mid-defoliation by significant enrichment, suggesting that WRKY may play an important role in regulating lemon leaf abscission. WRKY transcription factors are mainly expressed in response to abscisic acid and ethylene during organ abscission (*Zhang et al., 2023*). For example, WRKY8 regulates abscisic acid and ethylene signaling pathways and mediates crosstalk between abscisic acid and ethylene signals during TMVcg-*Arabidopsis* interactions, thereby conferring TMV-cg resistance (*Chen et al., 2013*). From this, it can be hypothesized that WPKY8 regulates ethylene and abscisic acid, which in turn regulates organ shedding (*Jiang et al., 2017*). In addition, WRKY transcription factors alter the transcript levels of related genes by activating salicylic acid (SA), JA, and ethylene signaling pathways (*Chen et al., 2012*). Therefore, it is hypothesized that the WRKY transcription factors identified in organ shedding could, in part, regulate organ shedding caused by the expression of related genes through the regulation of signaling pathways (*Wani et al., 2021*).

## CONCLUSION

This study used transcriptome analysis to explore the physiological indicators of two lemon varieties at three different leaf shedding stages. The results showed that the main physiological cause of leaf shedding was the accumulation of hydrolytic enzymes (cellulase and pectinase) as well as SOD and MDA activities. Comparative transcriptomics further identified a number of down-regulated DEGs involved in the IAA phytohormone synthesis pathway, as well as up-regulated DEGs in the β-glucosidase, peroxidase, and

lipoxygenase metabolic pathways. The results of physiological and transcriptomic analyses combined with WGCNA analysis resulted in a final screen of eight genes (*β-glucosidase* (*CL4G050040012_alt*, *CL7G062112012_alt*), *AUX/IAA* (*CL9G066930012_alt*), *SAUR* (*CL0G069923012_alt*), *GH3* (*CL2G041491012_alt*), *POD* (*CL5G056281012_alt*), and *WRKY* (*CL6G058882012_alt*, *CL1G038862012_alt*)) that are potentially involved in the lemon leaf abscission response. This study enriches the studies related to lemon leaf abscission and provides a theoretical basis for the subsequent validation of defoliation genes and analysis of the mechanism of lemon defoliation.

### Funding

This research was funded by National Natural Science Foundation of China (No. 31960574); Talent Revitalization Program(2022RC001). The funders had no role in study design, data collection and analysis, decision to publish, or preparation of the manuscript.

### Grant Disclosures

The following grant information was disclosed by the authors:
National Natural Science Foundation of China: No. 31960574.
Talent Revitalization Program: 2022RC001.

### Competing Interests

The authors declare that they have no competing interests.

### Author Contributions

- Meichao Dong conceived and designed the experiments, analyzed the data, prepared figures and/or tables, authored or reviewed drafts of the article, and approved the final draft.
- Tuo Yin analyzed the data, prepared figures and/or tables, and approved the final draft.
- Junyan Gao performed the experiments, prepared figures and/or tables, and approved the final draft.
- Hanyao Zhang analyzed the data, authored or reviewed drafts of the article, and approved the final draft.
- Fan Yang performed the experiments, prepared figures and/or tables, and approved the final draft.
- Shaohua Wang performed the experiments, prepared figures and/or tables, and approved the final draft.
- Chunrui Long performed the experiments, prepared figures and/or tables, and approved the final draft.
- Xiaomeng Fu performed the experiments, prepared figures and/or tables, and approved the final draft.
- Hongming Liu performed the experiments, prepared figures and/or tables, and approved the final draft.

**PeerJ** ______________________________

- Lina Guo performed the experiments, prepared figures and/or tables, and approved the final draft.
- Dongguo Zhou analyzed the data, prepared figures and/or tables, and approved the final draft.

## Data Availability

The raw sequence reads are available at GenBank: PRJNA1033435.

## Supplemental Information

Supplemental information for this article can be found online at http://dx.doi.org/10.7717/peerj.17218#supplemental-information.

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
