# Peer review of "Transcriptome differential expression analysis of defoliation of two different lemon varieties"

_PeerJ, doi:10.7717/peerj.17218_

## Round 0.1 · original submission · Major Revisions

Please revise the article considering the reviewers' comments. Also, take proper care of language and English.

**Language Note:** The Academic Editor has identified that the English language must be improved. PeerJ can provide language editing services - please contact us at copyediting@peerj.com for pricing (be sure to provide your manuscript number and title). Alternatively, you should make your own arrangements to improve the language quality and provide details in your response letter. – PeerJ Staff

·

Basic reporting

Overall study is good but presentation of work needs more clarity.

Experimental design

Appropriate

Validity of the findings

No comments

Additional comments

1. Figure 7- there are many number of DEG to find out more suitable contrasting genes between these two varieties stringent the log2 fold change value and top 15 genes you must express in final figure 7 and present figure go as supplementary figure.
2. Figure 9, 10 and 11 also look as supplementary figures.
3. Conclusion part so lengthy results should not come here major significant finding with future prospects must be there without numbering.
4. Figure 9 caption check the spelling.
5. Botanical name throughout manuscript needs to be italic.
6. Language :thorough corrections needed.
7. References 2023 must be added there are many research on DEGs.

Reviewer 2 ·

Basic reporting

This paper investigates the physiological indexes differences and transcriptome analyses between the two types of rootstock-scion combinations with focusing on defoliation trait in Lemon: 'Yunning No.1'/Poncirus trifoliate (L.) Raf and Allen Eureka'/Poncirus trifoliate (L.) Raf. This research is unique and special since it is not based on traditional samples from one individual. The samples are from the grafting process which may show a new scenario regarding the important breeding traits affected by the interference by the importing of physiological metabolites and nutrients from other individuals/varieties. The whole design is clear and reasonable. However, I am wondering whether the two varieties originally show differences in defoliation trait, and the author should introduce why defoliation trait is important in lemon breeding. Additionally, the author should also briefly introduce the genome analysis advance in Lemon since the main analysis is based on the genome data.
The author should also improve their writing by considering the organization of language, accuracy of presentation, and introduction to knowledge backgrounds outside this research field. There are a lot of essential citations missing in the introduction as well.

Experimental design

The experiment design is generally reasonable and solid. One thing is that the author missed the data from the three-year observation records. Additionally, there are a lot of details that were missing in the Materials and Methods section.

Validity of the findings

The validity of the findings is generally good.

Additional comments

Major revision:
1. Line95-96, “In the previous period, our group conducted observation records and research on two lemon varieties ('Allen Eureka' and 'Yunning No.1') for three consecutive years.”, the author leave a sentence here and didn’t show a detail of this result. The following description were talking about the comparison of rootstock combinations. However, the author still didn’t mention the three consecutive years comparation results between 'Allen Eureka' and 'Yunning No.1'.
2. The whole introduction should be revised significantly: organization, process details of many operation (the author should offer more photos or make a graph of clearly showing designing of these field experiments), biochemical measure details (“A kit was used”, “repeated three times” sample repeats or biological repeats?, “MD SpectraMax 190 full-wavelength enzyme labeler” what are the basic parameters? How does author prepare the samples for the enzyme labeler? How many samples amounts were loaded? ). More details should be supplied in 2.4 Hormone Determination, 2.5 RNA extraction, RNA-seq library preparation, and transcriptome sequencing, and 2.6 Transcriptome data analysis
3. The author conducted both physiological indexes and transcriptome analyses, however, they did not mention the physiological indexes results in the abstract and didn’t discuss the connection between these two results.
4. The author should also compare their 11 figures with their Materials & Methods section to see what kind of Methods they were missing. For example, how do they make these heatmaps?
5. Figure10 tell the reader nothing.
6. what is the value of the right side of the Figure 9?
7. How do the author make the Figure 8?what do the different modules mean?
8. The gene names of Figure 7 is vague.
9. Table1 should be in the supplementary data. Some figures should be moved to the supplementary data as well.


Minor revision:
1. In Abstract: “the predefoliation stage (CQ), the middefoliation stage (CZ), and the postdefoliation stage (CH).”, what do CQ CZ and CH really mean? “WGCNA”, what does this mean? can you use full name? “AUX/IAA, SAUR, GH3, POD, WRKY, AP2, and bZIP”, are these specific genes names? Or just gene family? “A total of 898, 4856, and 3126 differentially expressed genes (DEGs) were obtained in CQ, CZ, and CH, respectively”, what kind of threshold values you use?

2. Line42, “It has the effect of lowering blood fat, uric acid, antioxidants, and cancer”, grammar error.

3. Line45, “Lemons are tropical and subtropical evergreen tree species.” This sentence should be move to the beginning of the paragraph.

4. Line45, “and subtropical evergreen tree species. However, the production of 'Allen Eureka' lemon was”, there should be having one sentence to connect these two sentences to introduce what is 'Allen Eureka' lemon, why you focus on this variety.

5. Lin48, “in the plant kingdom”, Abscission is a widespread physiological process in animal as well.

6. Line49, “is very similar”, what does it exactly mean? Conserved? Similar morphology?

7. Line48-49, “Abscission is a widespread physiological process in the plant kingdom, including leaves, flowers, fruits, roots, etc., and the mechanism of abscission is very similar in different organs.” Please cite references as well as many other sentence in the introduction.

8. Line50, “Abscission Zone”, should be “abscission zone (AZ)”,
9. Line52, “Arabidopsis thaliana” should be italic.
10. Line56-66, please cite papers fully.
11.Line74-79, “For example, in citrus, evergreen citrus was treated with ethylene glycol, transcriptome sequencing was performed on the treated abscission layer sites, and analysis of the RNA-Seq results showed that cell wall modifying enzyme- encoding genes, stress-related genes, pathogen-related genes, MAPK kinase-related genes, and transcription factors, as well as ethylene biosynthesis and signaling genes, were significantly expressed in the abscission layer locations [13-14].”this sentence is too long to read.
12. Line85-86, “To study the molecular regulatory mechanism of lemon defoliation”, this research is not a classical molecular regulatory mechanism research. Please rephase this statement.
13. Line85-92, this should be a short description of what and how do this study conduct and what is the conclusion and significance of this study. And should be a single paragraph.
14. Lines 86-91, explicitly state the knowledge gap that your study aims to fill. Highlight how your research contributes to the existing knowledge. This could include the uniqueness of studying two lemon varieties with different defoliation traits.
15. Line96-100, “the rootstocks were selected from Ziyang fragrant citrus, sour pomelo, red lemon, Valencia orange (nondeciduous rootstock in winter), and Poncirus trifoliate (L.) Raf (deciduous rootstock in winter).” should be “the five rootstocks were selected from Ziyang fragrant citrus, sour pomelo, red lemon, Valencia orange (nondeciduous rootstock in winter), and Poncirus trifoliate (L.) Raf (deciduous rootstock in winter), respectively.”
16. Line97, “for 'Allen Eureka'” should be “with 'Allen Eureka' scions”.
17. Line98, what is the specific name of the “sour pomelo, red lemon”?
18. Line100, remove “At the same time”.
19. Line101, what does “impact greater” mean?
20. Line96, “Five rootstocks” should be “Five rootstock types”, accordingly, Line100 , “with each rootstock” should be “with each rootstock type”.
21. Line102, “Eureka' growth potential.” should be “Eureka' growth.” “The 'Allen Eureka'/Ziyang fragrant citrus” should be “The 'Allen Eureka'-scion/Ziyang fragrant citrus-rootstock”.
22. Line101, “A year later”, the author should indicate the very detail growth condition for this one year.
23.Line100, the author should be better to show some figure about this process to extend the interest of their readers.
24. Line106, “was also observed in all five rootstocks grafted to 'Allen Eureka'”, “was” should be “were”,
25. Line 517, serial number is missing.

·

Basic reporting

Abstract:
1) Consider starting the abstract by providing a brief background on the importance of understanding leaf abscission in lemon cultivars.
2) Clearly state the objective or hypothesis of the study at the beginning to provide context. The abstract ends abruptly.
3) Consider adding a sentence or two to summarize the main findings and their significance.

introductuction:
4) The introduction is somewhat lengthy and covers a wide range of topics. Consider breaking it down into smaller sections to enhance readability and facilitate understanding.
5) Some points are repeated, such as the information about 'Allen Eureka' having abnormal defoliation in winter, which is mentioned multiple times in different ways. Streamlining the content and avoiding repetition can improve clarity.
6) Abbreviations such as "CZ" and "CH" are used without being defined. It's crucial to provide definitions for these abbreviations or include a list of abbreviations for clarity.
7) Some sentences are complex and could be simplified for better comprehension. Additionally, there are instances of awkward phrasing that could be improved for smoother reading.
8) The citation format at the end of certain statements (e.g., [5-12]) is inconsistent. Ensure a consistent and clear citation format throughout the text.
Material and Methods
9) The method describes comparisons between different rootstock combinations and varieties, but there is no mention of a control group. A proper control group is essential to isolate the effects of the experimental variables.
10) The method mentions three biological replicates for the RNA-seq analysis, but it is not clear how many technical replicates were performed for each sample. The number of replicates is crucial for assessing the robustness and reliability of the results.
11) The rationale behind choosing specific time points for sampling (November 15, 2021, December 1, 2021, and December 15, 2021) is not explained. Clarifying the decision-making process for these sampling dates and providing more context on the phenological stages would enhance the experimental design.
Results:
12)The section discussing changes in enzyme activities lacks interpretation or discussion of the biological significance of the observed changes. Providing explanations for the variations in enzyme activities during different defoliation periods would enhance the understanding of the results
13) The section briefly mentions the statistics of off-region transcriptome sequencing data, but it lacks information on the criteria used for data quality control and normalization. Providing more details on these aspects would enhance the reliability of the transcriptome data.
14)The description of gene coexpression networks is complex, and the significance of identified hub genes is not clearly explained. A more straightforward presentation of the key findings and their implications would improve reader comprehension.
15) While the number of differentially expressed genes is presented, there is a lack of discussion regarding the biological relevance of these genes in the context of defoliation. Providing insights into the potential functions of the identified genes would help readers better understand their significance.
Discussion:
16)The discussion lacks a clear synthesis of findings and their broader implications. It would benefit from connecting the individual results to the overall research question and objectives. The author should emphasize how each finding contributes to the understanding of fruit tree defoliation.
17)While various findings are presented, some are not adequately explained. For example, the significant differences in metabolic activities in the mid-defoliation stage are mentioned, but the specific metabolic pathways and their implications are not discussed in detail. Providing more context and interpretation for key findings would enhance the discussion.
18) While the study mentions the WGCNA analysis and identifies certain transcription factors (WRKY, AP2, bZIP) as significantly upregulated in the blue module, the significance of these findings is not thoroughly explained. A more detailed discussion on the potential roles of these transcription factors in the context of leaf abscission regulation is needed.
19) The discussion lacks references to existing literature or comparative analysis with other studies on fruit tree defoliation. Providing comparisons with similar studies or discussing how the findings align with or deviate from existing knowledge would strengthen the argument and place the study in a broader context.

Experimental design

The objectives of the study are clearly stated, focusing on understanding the molecular regulatory mechanisms of leaf abscission in lemon varieties. The identification of differentially expressed genes (DEGs) in each defoliation stage and the focus on the largest number of DEGs in the middefoliation stage (CZ) suggests a thorough exploration of gene expression dynamics. The combination of differentially expressed gene clustering analysis and Weighted Gene Co-expression Network Analysis (WGCNA) adds depth to the study, allowing for the identification of key genes and potential regulatory networks.
Overall, the experimental design appears robust, and t provides a concise overview of the study's objectives, methods, and key findings.

Validity of the findings

the study seems to have a well-structured and comprehensive experimental design.

Additional comments

Dear Editor,
I have thoroughly reviewed the manuscript that explores defoliation traits in 'Allen Eureka' and 'Yunning No. 1' lemon varieties for gene discovery. The authors have made an excellent choice for their gene discovery approach. However, I would like to propose the following suggestions for consideration, aimed at enhancing the overall quality of the manuscript:

Grammatical and Formatting Issues:
The manuscript requires a careful proofreading to correct grammatical mistakes and ensure consistent spacing between words. A thorough edit is recommended for improved clarity and readability.
Sampling Clarification:

In the Materials & Methods section, please provide a clearer explanation of how sampling was conducted in different zones. This will help readers better understand the sampling procedures.
Enhancement of Figure 1:

Figure 1, illustrating various parameters for the defoliation stage, could benefit from the addition of a phenotypic figure. This addition will enhance the visual representation and overall comprehension of the figure.
Volcano Plot for Data Representation:

Consider creating a volcano plot to represent data, highlighting essential genes. This approach can streamline data presentation, making it more concise and easier for readers to follow.
Clarification on Experimental Design:

In the Methods section, clarify the number of independent biological samples and replicates used for both sequencing and qRT-PCR experiments. Providing this information will improve the transparency of the experimental design.
Exploration of Additional Genes:
It would be beneficial if the authors could study a few more genes involved in the defoliation of lemon leaves, further enriching the study.
Comprehensive Presentation of Figure 11:
Accompany Figure 11 with straightforward qPCR data in addition to the currently presented correlation graph. This will provide a more comprehensive view of the data.
Inclusion of Analysis of Variance (ANOVA):
Add an Analysis of Variance (ANOVA) to the bar chart in Figure 11 to strengthen the statistical analysis and interpretation.
Strengthening the Discussion:
The Discussion section requires additional depth. Please consider rewriting it to provide a more robust and insightful analysis of the findings.
I trust these suggestions will contribute to the refinement of the manuscript. Thank you for considering these recommendations. I look forward to the improved version of the manuscript.

---

## Round 0.2 · accepted · Accept

The article was approved by the reviewer and hence accepted for publication.

·

Basic reporting

The revised manuscript is well improved and I have no further suggestions and believe it is now good enough to be accepted in your valuable journal.
Professional English used throughout, clear and unambiguous.
Bibliographical references, sufficient background/context given.
Closed with relevant results for the hypothesis.

Experimental design

The revised manuscript is well improved and I have no further suggestions and believe it is now good enough to be accepted in your valuable journal.
The research question is clear, relevant and significant. Explains how the research addresses an identified gap in knowledge.
Consistent research conducted to a high technical & ethical standard.
Revised methods described with sufficient detail & information for replication

Validity of the findings

The revised manuscript is well improved and I have no further suggestions and believe it is now good enough to be accepted in your valuable journal.